# New Perspectives on the Old Uses of Traditional Medicinal and Edible Herbs: Extract and Spent Material of *Persicaria hydropiper* (L.) Delarbre

**DOI:** 10.3390/nu16193368

**Published:** 2024-10-03

**Authors:** Marina Jovanović, Jovana Vunduk, Dragana Mitić-Ćulafić, Emilija Svirčev, Petar Vojvodić, Nina Tomić, Laksmi Nurul Ismi, Dina Tenji

**Affiliations:** 1Institute of General and Physical Chemistry, Studentski trg 12/V, 11158 Belgrade, Serbia; jvunduk@iofh.bg.ac.rs; 2Faculty of Biology, University of Belgrade, Studentski trg 16, 11000 Belgrade, Serbia; mdragana@bio.bg.ac.rs (D.M.-Ć.); laksminrlism@gmail.com (L.N.I.); 3Faculty of Sciences, University of Novi Sad, Trg Dositeja Obradovića 3, 21000 Novi Sad, Serbia; emilija.svircev@dh.uns.ac.rs (E.S.); dinatenji@gmail.com (D.T.); 4Private Psychiatric Practice Psihocentrala Belgrade, Crnogorska 2, 11000 Belgrade, Serbia; petar.vojvodic@gmail.com; 5Group for Biomedical Engineering and Nanobiotechnology, Institute of Technical Sciences of SASA, Knez Mihailova 35/IV, 11000 Belgrade, Serbia; nina.tomic@itn.sanu.ac.rs

**Keywords:** spent plant material, post-extraction solid waste, *Persicaria hydropiper* (L.) Delarbre, psychobiotics, safety evaluation, zero-waste

## Abstract

**Background/Objectives:***Persicaria hydropiper* (L.) Delarbre, commonly known as water pepper, possesses multifunctional potential. Our research focuses on its complex phenolic composition, bioactivity, safety evaluation and utilization in a sustainable manner. Moreover, a survey was conducted among the Serbian population to gain insight into the attitude towards traditional wild-growing herbs (i.e., *P. hydropiper*), the level of familiarity with their zero-waste culture, and to assess eating behaviors. **Methods:** A survey was conducted with 168 participants to assess attitudes towards traditional herbs, zero-waste culture, and eating behaviors, while cytotoxicity, in vivo toxicity, chemical analysis of secondary metabolites, and probiotic viability assays were performed to evaluate the effects of the PH extract. **Results:** Notably, *P. hydropiper* extract (PH) exhibits a diverse phenolic profile, including quinic acid (3.68 ± 0.37 mg/g DW), gallic acid (1.16 ± 0.10 mg/g DW), quercetin (2.34 ± 0.70 mg/g DW) and kaempferol-3-*O*-glucoside (4.18 ± 0.17 mg/g DW). These bioactive compounds have been linked to anticancer effects. The tested extract demonstrated a cytotoxic effect on the human neuroblastoma cell line, opening questions for the further exploration of its mechanisms for potential therapeutic applications. Based on the toxicity assessment in the *Artemia salina* model, the PH could be characterized with good safety, especially for the lower concentrations (LC_50_ = 0.83 mg/mL, 24 h). The utilization of the spent PH material supports the viability of psychobiotic strains (up to 9.26 ± 0.54 log CFU/mL). Based on the conducted survey, 63.7% (*n* = 107) of respondents mainly prefer traditional instead of imported herbs. The respondents were skeptical about zero-waste edibles; 51.2% (*n* = 86) would not try them, and a bit more than half were not familiar with zero-waste culture (57.7%; *n* = 97). Only 8.3% (*n* = 14) followed a flexitarian diet as a dietary pattern. **Conclusions:** The use of underutilized traditional plants and their spent material could potentially contribute to the acceptance of a zero-waste culture in Serbia. Reinventing the use of neglected traditional plants and addressing ways for spent material valorization could contribute to the acceptance of a zero-waste strategy and encourage healthier eating behavior.

## 1. Introduction

Various agricultural residues, such as fruit and vegetable pomace, brewers’ spent grain, wine pomace, molasses, spent coffee grounds, spent tea leaves and post-extraction residues from both plant and fungal material, offer unexplored potential for sustainable utilization [1,2,3,4,5]. Simultaneously, the promotion of homemade plant-based meals aligns with zero-waste culture and emphasizes the reduction of by-products, specifically bio–agro waste. However, this food-from-waste concept, with dominantly vegetarian sources of macronutrients, encounters a cold welcome among Balkan countries. Interestingly, less than a century ago, the prevailing diet in the Balkan region centered around plant-based foods, particularly grains [6]. Historical dietary patterns also incorporated seasonally available domesticated plants, wild herbs and game meat. These culinary choices appear to align with sustainability, emphasizing locally sourced ingredients. However, the contemporary Serbian diet now mirrors the prevailing trend in many Western countries, characterized by dishes rich in saturated fat, sugar and sodium [7]. Current research, backed up by reports issued by health officials, point to a diet crisis. Proposed solutions for environment stability and healthier human nutrition [8,9] are directed towards repurposing bio–agro waste into sustainable foods as well as incorporating dietary supplements and less restrictive dietary patterns, such as the flexitarian diet [10,11,12]. It becomes imperative to expand the availability of studies focusing on best practice models related to plant-based meals and foods fortified with bio–agro waste. Rediscovering the edible and medicinal plants of the Balkan region, which were once integral to culinary heritage and served as home remedies, warrants attention. Exploring historical cookbooks’ recipes that incorporate plant-based meal preparations and wild-growing herbs, such as *Periscaria hydropiper*, *Persicaria bistorta*, *Asphodellus albus* and *Sempervivum tectorum*, could diversify meal options and facilitate the adoption of a flexitarian diet. Additionally, advocating for the utilization of ingredients such as spent material from traditional region-specific plants can enhance consumer trust and promote the acceptance of zero-waste food practices.

*Persicaria hydropiper* (L.) Delarbre, commonly known as water pepper (referred to as “papreni lisac” in the Serbian language), is a weedy yet health-promoting plant within the Polygonoideae subfamily. Widely distributed in the wild, it is an accessible reservoir of biologically active compounds and plant fiber [13]. Notably, *P. hydropiper* holds potential as both a natural remedy and a fiber-enriched supplement in foods. Its aromatic scent, along with its peppery, pungent and slightly bitter taste, renders it a viable culinary ingredient [14]. The leaves find application as vegetables in soups, stews, and salads, while young shoots are used for savory pies [15,16,17]. 

According to the extensive literature, ethnopharmacology and contemporary medicine practice, *P. hydropiper* has a record of antipathogenic, anticancer and immunomodulatory effects, promoting beneficial gut microbiota growth and brain functioning [13,18,19,20,21,22]. In more detail, recent publications focus on the influence of *P. hydropiper* in achieving better mental health, stating that its extracts showed good anxiolytic, anti-depressant, and sedative activity [20,21], while the essential oils derived from *P. hydropiper* could be employed in the treatment of neurodegenerative disorders such as Alzheimer [22,23,24]. Although a significant number of research indicates the potency of *P. hydropiper* in the treatment of mental and neurodegenerative disorders, they are all based on the use of extracts or essential oil [24]. 

There are two important trends when it comes to herb utilization: the global herbal extract market is expected to grow in the next five years, with a compound annual growth rate (CAGR) of 6.63% and the rising trend of homemade herbal preparations including extracts [25]. This trend opens the possibility to encourage more sustainable extract preparation and usage practices as well as to emphasize the importance of the further processing of plant spent material in order to obtain the culinary ingredient.

The potency of the spent plant material remaining after extraction has not been examined. Data regarding the remaining spent material influence on probiotic strains, which can also exhibit a psychobiotic effect, is scarce. The mentioned type of probiotic bacteria was named “psychobiotic”, while today this term refers to the potential of probiotics, but also paraprobiotics, prebiotics, and all microbiota-oriented interventions that can manipulate the microbiota-gut-brain axis (MGB) and have positive effects on neurological functions such as mood, cognition and anxiety, and thus produce health benefits in patients suffering from a psychiatric illness [26]. The viability of health-promoting microbes including psychobiotics, could be promoted by employing bio–agro waste as functional ingredients. 

This study aimed to explore *P. hydropiper*’s extract and its residues. The phenolic composition, toxicity and biological potential of *P. hydropiper* extract were estimated. Next, the effect of spent *P. hydropiper* material on psychobiotic bacteria viability was tested. Finally, a survey was conducted among the Serbian population to gain insight into the attitude towards traditional wild-growing herbs use, the state of knowledge about zero-waste culture, and to assess eating behaviors. Once known in the traditional cuisine and ethnopharmacology *P. hydropiper* offers a convenient bridge between zero-waste culture and re-adoption of a healthier eating behavior. 

## 2. Materials and Methods

### 2.1. Materials

The strains used for the in vitro antibacterial activity testing are as follows: *Escherichia coli* ATCC 8739, *Pseudomonas aeruginosa* ATCC 15442, *Shigella flexneri* ATCC 9199, *Staphylococcus aureus* ATCC 25923, *Salmonella* Enteritidis ATCC 13076, *Enterococcus faecalis* ATCC 29212, *Listeria monocytogenes* ATCC 19111 and probiotic strains (*Lactiplantibacillus plantarum* 299V, *Limosilactobacillus reuteri* DSM 17938, *Heyndrickxia coagulans* and *Escherichia coli* Nissle 1917), which were obtained from the Culture Collection of the Department of Microbiology, University of Belgrade, Faculty of Biology, Belgrade, Serbia. MRC-5 (ECACC 84101801) human cell fibroblast was obtained from the Oncology Institute of Vojvodina, Sremska Kamenica, Serbia. SH-SY5Y cell line (European Collection of Authenticated Cell Cultures, ECACC 94030304) was obtained from the Institute of Medical and Clinical Biochemistry, University of Belgrade, Faculty of Medicine. *Artemia salina* eggs were from Brine Shrimp Direct (Ogden, UT, United States [USA]) and NaCl was from Centrohem (Stara Pazova, Serbia). De Man–Rogosa–Sharpe broth (MRS broth) and MRS agar were obtained from Torlak (Belgrade, Serbia). Müller–Hinton broth (MHB), and brain–heart infusion (BHI) were obtained from HiMedia (Mumbai, India). Dulbecco’s modified Eagle’s medium (DMEM), Minimum Essential Eagle (MEM), Nutrient Mixture F-12 Ham (Ham’s F12), phosphate-buffered saline (PBS), dimethyl sulfoxide (DMSO), penicillin-streptomycin mixtures, trypsin from the porcine pancreas, sodium pyruvate, HEPES and 3-(4,5-dimethylthiazol-2-yl)-2,5-diphenyltetrazolium bromide (MTT) were purchased from Sigma-Aldrich (Steinheim, Germany). Fetal bovine serum was purchased from Gibco (Grand Island, NY, USA) and NaHCO_3_ from Roth (Karlsruhe, Germany). The reference standards of the secondary metabolites were obtained from ChromaDex (Santa Ana, CA, USA), Fluka Chemie gmbh (Buchs, Switzerland) or Sigma-Aldrich Chem (Steinheim, Germany). HPLC gradient grade methanol was purchased from J. T. Baker (Deventer, The Netherlands), and p.a. formic acid and DMSO from Merck (Darmstadt, Germany).

### 2.2. A Survey Regarding the Traditional Herbs Usability, Zero-Waste Culture and Eating Behaviors

A survey regarding the modern consumer attitudes towards traditional herbs, zero-waste culture and eating behaviors was conducted as described by [27]. A survey was conducted using a web-based questionnaire. The link to the open survey was shared (Google form) by a private psychiatric practice Psihocentrala (Belgrade, Serbia) to clients, who acted responsibly in regard to all aspects of their health and life choices. The link was shared through the internal mailing list. One response per participant was allowed. The data collection ran for one month. The study began in April 2023 and was completed in May 2023. The condition of participation in the research was the age limit; the participants had to be 18 years or older. The questionnaire contained 21 closed questions about the survey participants’ personal data (age, sex, birthplace), the prevalence of traditional herbs usage and their level of familiarity with the zero-waste culture and eating behaviors (food-related behaviors, interest in plant-based food, attitude and willingness to try different types of food). It is important to note that participation in this study was entirely on a voluntary basis. The participants were informed about the purpose of the survey, the research team involved, and the time required to complete the questionnaire. A total of 168 responses were collected. In more detail, a total of 168 respondents participated [52.4% (*n* = 88) female, 47.0% (*n* = 79) male and 0.6% (*n* = 1) respondents did not identify with either gender]. The majority of the respondents were between 30 and 45 years old [30–35 years, 17.9% (*n* = 30); 35-40 years, 30.4% (*n* = 51); 40–45 years, 26.2% (*n* = 44)]. Most of the respondents declared that they were born in Serbia (88.1% (*n* = 148)). The sample size was determined using the Raosoft online calculator (Maple Tech. International LLC, TX, USA). A minimum of 162 participants was necessary to achieve 95% statistical power and a significance level of 5% (*p* = 0.05). The survey is available in its translated form as Appendix A. Surveying with the modern customers including psychiatric users was approved by the local Ethical Committee at Ćuprija General Hospital, Serbia (No. 10332/1/4 decision made on 12 September 2023)). The participation of the respondents in the survey was in line with the Code of Professional Ethics of the University of Belgrade (Senate of the University of Belgrade, 2016), Official Gazette of the Republic of Serbia, 189/16, 16. 

### 2.3. Extract Preparation 

Dry aerial parts of *P. hydropiper* were provided in the form of tea leaves from a certified herbal supplier (Belgrade, Serbia). First, the plant material was grounded (Ariete GrinderPRO, Italy) to fine powder. Extracts were prepared by macerating powdered plant material (10 g) in a 50% water/ethanol solution (100 mL, mass:solvent ratio 1:10) for 24 h with constant stirring (150 rpm) at room temperature. Then, the liquid was separated from the plant material by filtration and centrifuged at 6200× *g* for 10 min (Universal 320 R, Hettich Zentrifugen, Germany). The supernatant was filtered again before collection and storage at 5 °C. The procedure was repeated two more times. The supernatants that collected for three subsequent days were combined and the solvent was evaporated using a rotary evaporator (RV 8 IKA, Staufen, Germany) at 39 °C under vacuum. The protocol used in this study was adjusted to enable a high level of material exhaustion and the final solid residue represented spent plant material. The dried extracts were dissolved in DMSO to a final concentration of 100 mg/mL. The spent PH material remaining after extraction was dried at room temperature for several days, until the complete disappearance of moisture, ventilating it periodically for 1h in a laminar chamber. The PH extract was subjected to tandem mass spectrometry (LC-MS/MS) analysis, MIC and MBC antimicrobial assays and in vivo and in vitro toxicity evaluation. The spent PH material was used in the assessment of the viability of probiotic bacteria.

### 2.4. Chemical Analysis

A quantitative liquid chromatography with a tandem mass spectrometry (LC-MS/MS) analysis of 45 selected secondary metabolites was performed according to a previously reported method [28] with a detailed procedure and validation. A standard mixture containing 45 phenolics was diluted with mobile phase solvents A (0.05% aqueous formic acid) and B (methanol) in a 1:1 ratio to create fourteen working standards ranging from 12,500 ng/mL to 1.53 ng/mL. The extracts were also diluted with solvents A and B (1:1) to a final concentration of 2 mg/mL. The samples and standards were analyzed using an Agilent Technologies 1200 Series high-performance liquid chromatograph coupled with an Agilent Technologies 6410A Triple Quad tandem mass spectrometer (Agilent Technologies, Inc. Santa Clara, CA, USA) with an electrospray ion source (ESI), controlled by the Agilent Technologies MassHunter Workstation software, Data Acquisition (version B.03.01).

### 2.5. Antibacterial Activity of the PH Extract

#### 2.5.1. Growth Conditions and Strains Used in Minimum Inhibitory Concentration (MIC) and Minimum Bactericidal Concentration (MBC) Assay 

The following Gram-negative *E. coli* (ATCC 8739), *P. aeruginosa* (ATCC 15442) *S. flexneri* (ATCC 9199), and *S.* Enteritidis (ATCC 13076) and Gram-positive bacterial strains were used in in vitro antimicrobial assays: *S. aureus* (ATCC 25923), *E. faecalis* (ATCC 29212) and *L. monocytogenes* (ATCC 19111). Bacterial overnight cultures were prepared in MHB broth, except *L. monocytogenes* cultures, which were grown in BHI.

#### 2.5.2. MIC and MBC Assay

The standard broth microdilution method recommended by the Clinical and Laboratory Standards Institute [29] was used to determine the antibacterial activity of the PH. Briefly, the overnight cultures of the tested strains were pelleted and resuspended in 0.01 M MgSO_4_ to reach a 10^6^ CFU/mL. The MIC assay was carried out in 96-well microtiter plates by preparing serial twofold dilutions of the tested substances (up to 5 mg/mL) in the appropriate media (MHA and MHB). To each 180 µL of the dilutions, 20 µL of bacterial suspension and the aqueous solution of resazurin (final concentration of 0.0675 mg/mL) were added to each well. After overnight incubation (18–20 h at 37 °C), the MIC values were determined as the lowest concentrations of the tested substances without a visible color change. The MBC values were determined by plating 10 µL of the samples from wells without visible growth onto the appropriate solid medium. For each strain, three independent experiments were performed in triplicate.

### 2.6. Cytotoxicity Assessment 

A cytotoxicity assay was carried out with PH by employing an MTT assay. Prior to the cytotoxicity test, the PH was sterilized (sterile syringe filter, 0.22 µm). Before cytotoxicity testing, both the cell lines were cultivated as a monolayer in a T-75 culture flask from Greiner Bio-One, and incubated at 37 °C in a humidified atmosphere containing 5% CO_2_. The flask was subcultured twice per week using the conventional trypsinization procedure. The SH-SY5Y cells were grown in a 1:1 mixture of Minimum Essential Eagle (MEM) and Nutrient Mixture F-12 Ham (Ham’s F12), which was supplemented with 10% fetal bovine serum (FBS,), 1% HEPES, 2.2 g/L NaHCO_3_, 1% penicillin-streptomycin solution and 0.055 g/L sodium pyruvate, while MRC-5 cells were grown in DMEM medium with 10% fetal bovine serum, 1% penicillin/streptomycin mixtures and 2 mM of L-glutamine. When the cell cultures reached 90% confluence, they were collected and placed into 96-well plates with a density of 2 × 10^4^ cells/well. After the overnight incubation, the cells were treated with diluted extracts at the following concentrations: 1; 0.75; 0.5 and 0.25 mg/mL, and incubated for 24 h. After that, the medium with test substances was replaced with MTT (final concentration 0.5 mg/mL). After 3 h the medium was removed and the formazan crystals were dissolved in DMSO. The cell viability was assessed by measuring the absorbance at 540 nm (A540) and 690 nm (A690) using a microplate reader (Multiskan FC, Thermo Scientific, Shanghai, China). The absorbances used for further calculations were obtained as A540–A690 nm. All the extracts were tested in hexaplicates in two independent experiments.

### 2.7. Safety Evaluation of PH Extract by Using Artemia salina Toxicity Assay

For the assessment of the in vivo toxicity of the extract, a *Artemia salina* brine shrimp toxicity assay was performed. *A. salina* is a commonly used and reliable model organism for determining the presence of biological activity and the preliminary toxicological testing of natural compounds [30]. 

The test procedure, modified slightly from Rajabi et al. [31], was as follows: brine shrimp eggs were incubated in salt water containing 35 g/L NaCl for 12 h, at 30 °C, under constant aeration and illumination. Hatched nauplii were collected and separated from shells and unhatched eggs 2 h after the beginning of hatching, ensuring the same age of the test organisms (Instar I) [32]. The groups of nauplii contained in the 150 μL of water (5 nauplii per well in the 96-well plates) were exposed to final concentrations of 0.25–1 mg/mL of PH, obtained by adding 50 μL of the according stock concentration, prepared by diluting the original stock in salt water. The plates were then incubated at the ambient temperature for 24 and 48 h, before counting the live organisms, by using DigiMicro 2.0. Scale digital microscope and AMCap software 9.1. Larvae were considered dead if no movement was observed during 15s. The survival was counted compared to the survival of the control group. The LC_50_ value was calculated via a Probit analysis [33]. Two independent experiments were performed.

### 2.8. Viability of Probiotic Bacteria with Psychobiotic Potential

The influence of spent PH material on the viability of probiotics with psychobiotic potential has been examined in G positive, G negative and spore-forming bacteria. Prior to the screening, all probiotic strains were cultivated in MRS broth for 48 h at 37 °C under microaerophilic conditions, except *E. coli* Nissle 1917, which was grown in MHB. Selected strains were added in the appropriate medium (MRS and MHB) with and without spent PH material (1%). The viability of the health-promoting strains was evaluated at 0 h and 4 h using the pour plate technique and serial dilutions in phosphate-buffered saline (1 × PBS). LABs were enumerated using appropriate agar (MRS and MHA) after incubation at 37 °C for colony growth. For each strain, two independent experiments were performed in triplicate. The results were expressed as the log of the mean number of the colony-forming units (log CFU/mL). 

### 2.9. Statistical Analysis

The following programs for statistical data analysis were combined and used: software GraphPad Prism 6.0 (GraphPad Software Inc., San Diego, CA, USA) and Excel 2016 (Microsoft). The data obtained from the MTT assay and probiotic bacteria viability were analyzed via an analysis of variance (one-way ANOVA, Dunnett multiple comparisons test). The data obtained from the in vivo toxicity assessment were analyzed via an analysis of variance [one-way ANOVA, Tukey’s honestly significant difference test (HSD)]. The level of statistical significance was defined as *p* < 0.05. The data collected from the nutritional questionnaire were transferred into a spreadsheet (Excel, Microsoft) and submitted for descriptive analysis.

## 3. Results and Discussion

### 3.1. Attitudes Regarding the Traditional Herbs’ Usability, Zero-Waste Culture and Eating Behaviors of Serbian Consumers

To gain an insight into the eating habits of the population in Serbia and the attitude towards the use of traditional herbs and their residues, a survey was conducted. The Serbian population is predominantly oriented towards traditional cuisine, which is also perceived as healthy and has a positive image [34]. The majority of respondents, 85.1% (*n* = 143) prefer an omnivorous diet (meat and dairy products are dominant), while only 8.3% (*n* = 14) of respondents follow a flexitarian diet as a dietary pattern. As stated by Zrnić et al. [35], among the best-known gastronomic products of Serbia are fine pork cracklings in a tobacco-like shape, locally called *duvan čvarci*, and thick and fermented clotted cream, known as *kajmak.* Both of these products are characterized by a high-fat content. Not only they are they among the favorite ones based on their taste, but they also represent a point of national pride, so called *gastronationalism* [36]. In line with this, 63.7% (*n* = 107) of respondents pointed out that they would rather try products containing indigenous herbs than products containing imported botanicals. Also, 91.1% (*n* = 153) of the participants confirmed that they use herbs in the form of infusion and extract. However, only 7.7% (*n* = 13) of respondents use *P. hydropiper* as a cooking ingredient, while 4.2% (*n* = 7) are aware that this herb could contribute to the treatment of neurological and mental disorders. 

More than half of the respondents (57.7%, *n* = 97) were not familiar with the meaning of zero-waste, and 51.2% (*n* = 86) would not try zero-waste edibles. To be more precise, 51.2% (*n* = 86) would not consider introducing the ingredient or product, based on spent plant material from infusion and extract preparation, into their regular diet. Not all nationalities are equally open to try new specialties [35]. According to Bogusz et al. [37], 69.4% (*n* = 503) of respondents from Poland, 71.0% (*n* = 138) of modern consumers in Ukraine, and 86.7% (*n* = 180) of the surveyed population of Slovakia are familiar with the zero-waste concept. So far, there are no data examining the level of zero-waste knowledge in Serbia. In our study, it has been shown that a significant percentage of modern consumers, including users with increased awareness and self-care for common mental health, exert skepticism towards novel kinds of food. In more detail, 35.1% (*n* = 59) are skeptical about choosing novel food, while 28.0% (*n* = 47) are hesitant to try food they have never tried before. Moreover, 51.8% (*n* = 87) of respondents pointed out that they will not try food with, for them, unknown preparation and complete composition, regardless of the reputation of the place where it is marketed. As reported by Tuorila and Hatrmann [38], unfamiliar ingredients and foods (e.g., cultivated meat, insects) are negatively perceived by consumers. They add that, on the other hand, familiarity with food reduces anxiety and the suspicion of the food, so products characterized by the word “traditional” have higher hedonic ratings. Egolf et al. [39] showed the difference in the level of disgust towards food between the genders. Female respondents exerted a slightly higher degree of skepticism and pickiness compared to the male respondents, which is in line with the results of our research (Table 1). Although European and Western affluent societies have a diverse selection of foods available, the basis of our food selection mechanisms was developed in times of more limited supply. Foraging new food sources by sampling different botanicals posed a possible risk of poisoning. The so-called Rozin’s omnivore’s dilemma is based on opposing aspirations. On one hand, there is an urge to look for new food to satisfy curiosity and avoid short-term sensory-specific satiety and on the other hand, there is a tendency to fear new foods, caused by neophobia. It is assumed that food neophobia as well as the disgust function originated in the prevention of the oral ingestion of toxic or offensive agents [39,40]. Moreover, food disgust is oriented primarily toward eating behavior that prevents risky choices, like food with a potentially high pathogen load and rotten food. 

Our further research was therefore directed towards the characterization of the traditional, overlooked and underutilized herb *P. hydropiper*, its extract and its residues. Research included phenolic composition, biological activities, including the effect on both pathogenic and probiotic bacteria, and the evaluation of in vitro and in vivo toxicity. 

### 3.2. Phenolic Profile of PH Extract

Available evidence supports that some dietary strategies, such as employing a dietary pattern that relies on (poly)phenols-rich plants, may delay, prevent or heal chronic illnesses by exerting anti-pathogenic properties [41,42] and inhibiting cancer cell growth. Our results showed that the PH is rich in phenolic acids and flavonoids (Table 2). Moreover, the extract had significant amounts of quinic (3.68 ± 0.37 mg/g DW) and gallic acid (1.16 ± 0.10 mg/g DW) as well as quercetin (2.34 ± 0.70 mg/g DW) and quercetin derivatives, including quercetin-3-*O*-glucoside (3.81 ± 0.11 mg/g DW) and quercetin-3-*O*-galactoside (2.07 ± 0.12 mg/g DW). Significant amounts of kaempferol (1.01 ± 0.07 mg/g DW) and kaempferol-3-*O*-glucoside (4.18 ± 0.17 mg/g DW) were also detected. Plants from the Polygonoideae subfamily are generally rich in phenolic acids and flavonoids. For example, seed extract from common buckwheat, *Fagopirum esculentum* L., contains quercetin-3-*O*-glucoside up to 2.25 ± 0.17 (mg/g DW) and quercetin-3-*O*-galactoside up to 2.02 ± 1.23 [43]. Similar to PH extract, herb extract from *Polygonum aviculare* L., knotweed, is characterized by a high content of quinic acid (8.72 ± 0.87 mg/g DW), kaempherol-3-*O*-glucoside (1.33 ± 0.05 mg/g DW), quercetin-3-*O*-glucoside (1.38 ± 0.04 mg/g DW) and quercetin-3-*O*-galactoside (3.02 ± 0.18 mg/g DW), but compared to PH *P. aviculare,* herb extract has a lower amount of gallic acid (0.95 ± 0.09 mg/g DW), quercetin (0.40 ± 0.12 mg/g DW) and kaempferol (0.12 ± 0.01 mg/g DW) [44]. For an additional comparison, overripe fruits of elderberries (*Sambucus nigra* L.) contained quercetin 3-*O*-glucoside (0.35 ± 0.18 mg/g DW), kaempferol 3-*O*-glucoside (0.07 ± 0.03 mg/g DW) and quercetin (0.01 ± 0.02 mg/g DW) [45].

The content of individual compounds and their forms appear as very important. For example, several studies examined quercetin bioavailability in connection with the source. Black tea contains a higher amount of quercetin than onions, but its bioavailability is lower [46,47]. As demonstrated by Hollman et al. [48], onions contain quercetin glucoside, characterized by the highest extent of intestinal absorption. The PH extract from our study contained a significant amount of quercetin-3-*O*-glucoside in particular, pointing to the potentially high dietary quality of this plant. Interestingly, it has been proven that the bioavailability of quercetin, through absorption in the small intestine, increases in the presence of fat [49]. Considering that Balkan (and Serbian diet) is rich in fat, the inclusion of *P. hydropiper* is even more justified. 

Another compound detected in the PH extract that is of particular importance for a modern diet is kaempferol. It has been suggested that its consumption in an amount above 1.5 mg/day may be associated with lower coronary heart disease mortality (CHD) [46]. CHD results from coronary artery disease (CAD) and is the foremost cause of mortality and the loss of years of full health [50]. It is connected with obesity, smoking, and an aging population, all factors with a significantly growing rate in Serbia [51,52]. The daily consumption of fruits and vegetables is among the modifiable risk factors for CAD [50]. In this context, the incorporation of *P. hydropiper* into contemporary meals presents a viable route for health improvement, while paying attention to sustainable dietary patterns.

### 3.3. Bioactivity of the PH Extract

According to the literature data, as far as the dietary flavonoids are concerned, PH dominant compounds gallic acid, quercetin-3-*O*-glucoside, quercetin-3-*O*-galactoside, kaempferol, and kaempferol-3-*O*-glucoside, might exert antibacterial properties and cytotoxic activity on human cancer cells [53,54,55]. Here, the antibacterial effect of the PH extract was tested against seven pathogenic strains (in the given range of concentrations; up to 5 mg/mL), but the antibacterial effect was lacking. The absence of antibacterial potential against *E. coli* is contrary to the literature data [56]. This discrepancy could be attributed to the difference in the process of extraction or sensitivity of the tested strain. 

Next, the cytotoxicity of the PH extract was tested on non-differentiated human neuroblastoma SH-SY5Y cells. As shown in Figure 1, 24 h treatment with the PH caused a dose-dependent reduction in SH-SY5Y cell viability, significantly for doses exceeding 0.25 mg/mL. Natural polyphenols such as apigenin and epigallocatechin gallate can reduce the growth of SH-SY5Y cells. Furthermore, quercetin and luteolin can exert cytotoxicity on the mouse NB cell line, Neuro-2 [57]. PH has a significant amount of quercetin (2.34 ± 0.70 mg/g DW) which can explain the sample’s ability to inhibit cell growth. Luteolin (0.29 ± 0.02), epigallocatechin gallate (0.14 ± 0.01) and apigenin (0.02 ± 0.00) were also detected and might be responsible for the observed effect.

### 3.4. Safety Evaluation of the PH Extract

Next, the assessment of the PH cytotoxicity in vitro in MRC-5 human cell fibroblast (Figure 2) and the evaluation of PH in vivo toxicity (Figure 3) in the *A. salina* model system were conducted. We found that PH ethanol extract does not decrease the viability of MRC-5 cells at a concentration lower than 0.5 mg/mL. Furthermore, it can be seen that after 24 h, the PH extract in concentrations up to 0.5 mg/mL did not induce a significant decline in the survival of *Artemia* larvae. However, higher mortality was observed at 1 mg/mL, and the survival after 48 h was lower at all tested concentrations, which is consistent with the very high sensitivity of the Instar II phase of *Artemia* larval development. For the 24 h test, LC_50_ was 0.83 mg/mL, and 0.35 mg/mL after 48 h. These values of LC_50_ indicate the absence of toxicity, or a low toxicity of the extract’s lower tested concentration [58,59]. 

There are few studies on the toxicology of *P. hydropiper*. At present, Kong et al. [60] discussed that the flavonoid extract of *P. hydropiper* (<5 g/kg/BW) had no acute toxic effects or side effects in mice and has no sub chronic toxic effects after long-term continuous medication. Also, Kuroiwa et al. [61] discussed that in the consumption of *P. hydropiper* extract in the amount of 57.4 and 62.9 mg/kg/day for male and female rats, adverse effects have not been observed. Moreover, in an acute toxicity test, Raihan et. [62] reported the absence of a lethal effect, in Swiss-Webster mice, of *P. hydropiper* methanol extract in the amount of 400 mg/kg. However, Ayaz et al. [63] reported significant acute toxicity in a brine shrimps sample with lethality ranging from 54.43% to 93.33% at a concentration of 0.25 mg/mL depending on the solvent used for extraction.

### 3.5. The Impact of Spent PH Material on Probiotic Bacteria Viability

Bio–agro waste, including spent plant material, could be suitable for obtaining functional foods, supplements and nutricosmetics [1]. The spent PH material (1%) supported the high viability of the following bacteria: *L. plantarum* 299V, *L. reuteri* DSM 17938, *H. coagulans* and *E. coli* Nissle 1917 (Table 3). Similar to our results, several bagasses (residues of seed, peel, pulp, or stem) from grape, goji, tamarind, blackberry, and blueberry could support the high viability of probiotics, such as *Lacticaseibacillus rhamnosus* GG [64]. Also, the buckwheat hulls could promote the growth of beneficial bacteria like *Lactobacillus bulgaricus* [65]. Numerous chemical compounds could positively affect the viability of health-promoting microbes. Traditionally carbohydrate-based prebiotics, but also substances like phenolic compounds and conjugated fatty acids, can additionally meet this function [64]. It has been confirmed that dietary quercetin, present in our sample in a significant amount, has prebiotic-like properties, thus altering gut microbiota [66,67]. Moreover, Bian et al. [68] investigated the effect of kaempferol supplementation on the gut microbiota of mice on a high fat diet. The authors concluded that kaempferol had a great impact on the gut microbiota composition which further counteracted the gut microbiota dysbiosis. This was possible due to the kaempferol prebiotic-like effect. In our study, although the high viability rate of probiotic strains was maintained in the presence of spent PH material, and even though a slightly higher viability has been detected, it was not statistically significant. Phenolic compounds like quercetin and kaempferol should be present in higher amount in order to significantly affect beneficial bacteria growth.

## 4. Conclusions

Neglected and underutilized, *P. hydropiper* can be characterized as a dietary (poly)phenols-rich plant with quinic acid, gallic acid, quercetin, quercetin-3-*O*-glucoside, quercetin-3-*O*-galactoside and kaempferol-3-*O*-glucoside being dominant compounds. *P. hydropiper* extract exhibited significant cytotoxicity against human neuroblastoma cancer cells. In parallel, based on the in vitro and in vivo toxicity assessment, extracts could be characterized with good safety, especially for lower concentrations. However, to unequivocally characterize PH extracts and spent material as safe for human consumption, additional testing is required. Psychobiotic viability remains high in the presence of spent PH material.

The Serbian population, as a part of the Balkan nations, is characterized by food skepticism and the reluctance to try new food. There is also a need to raise awareness about the leading concepts of sustainable food practices, like zero-waste. Traditional plants represent an indirect and smart way to overcome alternative and novel food skepticism, and introduce novel eating concepts and encourage heathier eating behavior. It should be kept in mind that a more detailed insight (including a wider population age range) into the interest of the Serbia and Balkan populations towards zero-waste food and eating behavior is needed. 

## Figures and Tables

**Figure 1 nutrients-16-03368-f001:**
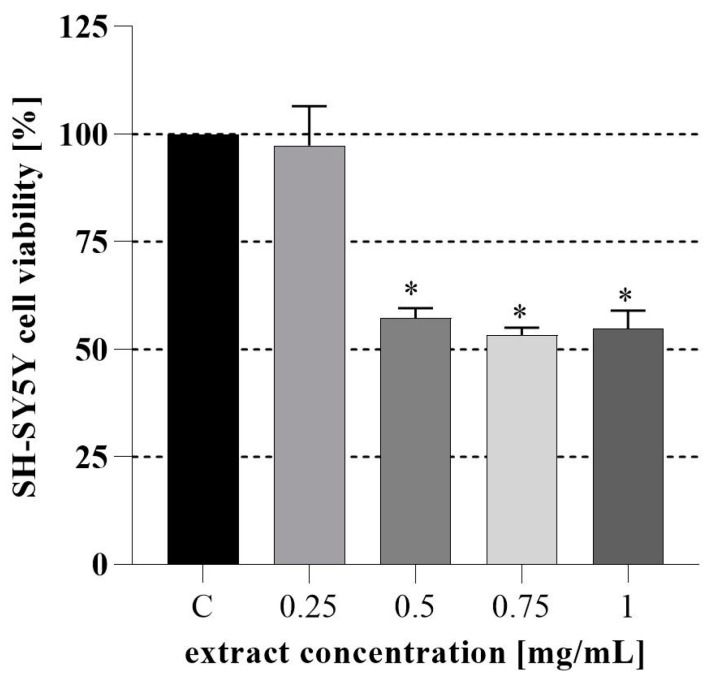
Inhibition rates of SH-SY5Y cells treated with PH after 24h; C-cell growth control. * A significant difference in means between all samples and C-control according to the Dunnett tests (*p* < 0.05).

**Figure 2 nutrients-16-03368-f002:**
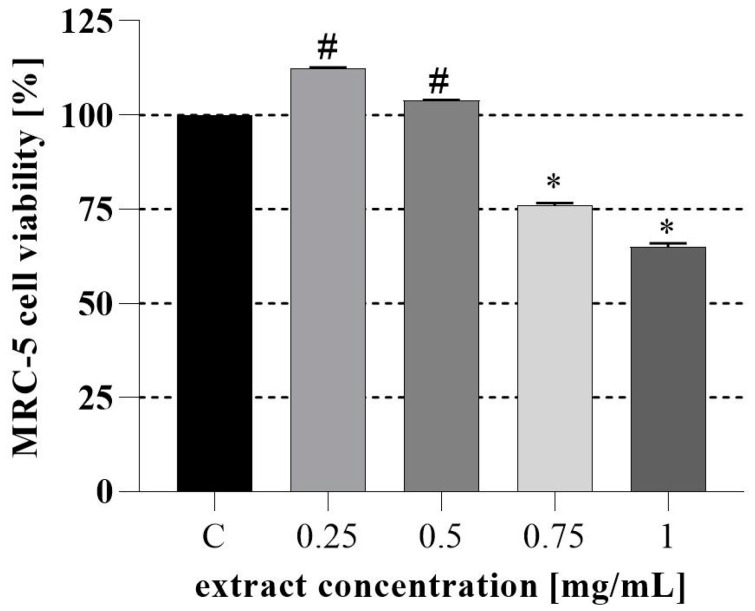
Assessment of PH cytotoxicity in vitro using human embryonic fibroblast MRC-5 cells. ^#^* significant difference between samples and GC (growth control) according to the Dunnett test (*p* < 0.05); ^#^ increased cell viability; * decreased cell viability.

**Figure 3 nutrients-16-03368-f003:**
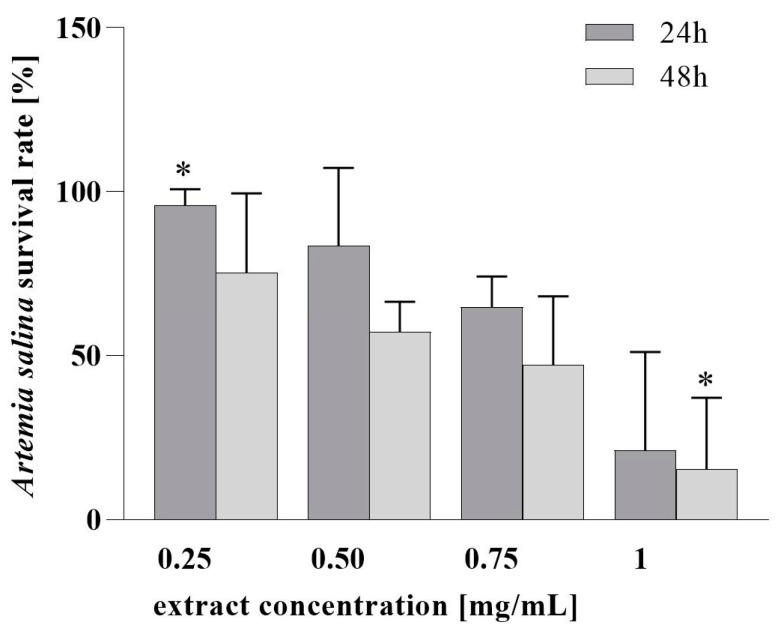
Assessment of PH toxicity in vitro in brine shrimps model. * A significant difference in means between samples according to HSD test (*p* < 0.05).

**Table 1 nutrients-16-03368-t001:** Familiarity with the zero-waste culture and eating behavior in relation to gender.

	[%] Female		[%] Male		[%] All Respondents in Specified Subgrop (Both Gender)
Subgroup	Relation to All Respondents (*n* = 168)	Relation to All Female Respondents (*n* = 88)	Relation to All Respondents (*n* = 168)	Relation to all male respondents (*n* = 79)	Relation to All Respondents (*n* = 168)
Familiarity with zero-waste culture (yes) ^a^	25.60	48.86	15.48	32.91	41.07
Familiarity with zero-waste culture (no) ^a^	27.98	53.41	29.76	63.29	57.74
Skepticism towards new types of food *	18.45	35.23	16.67	35.44	35.12
Reaching out for unexplored food *	33.93	64.77	30.36	64.56	64.29
Tasting food with unknown ingredients in a familiar restaurant (yes) *	25.00	47.73	22.62	48.10	47.62
Tasting food with unknown ingredients in a familiar restaurant (no) *	27.38	52.27	24.40	51.90	51.78
Fobia and pickiness of unfamiliar food *	14.29	27.27	13.69	29.11	27.98
Aspiration towards the food of different cultures *	38.10	72.73	33.33	70.89	71.43

^a^ Two respondents refused to answer the question; * one respondent did not state his attitude. All respondents *n* = 168; all female respondents *n* = 88; all male respondents *n* = 79, one respondent did not identify with either gender.

**Table 2 nutrients-16-03368-t002:** Determined content of selected secondary metabolites in PH.

Class of Secondary Metabolites	Compounds	Content [μg/g DW] *
**Cyclohexanecarboxylic acid**	**Quinic acid**	**3680 ± 368.0**
**Hydroxybenzoic acids**	**Gallic acid**	**1155 ± 104.0**
	Protocatechuic acid	198 ± 16.0
	2,5-dihydroxybenzoic acid	16.7 ± 1.3
	*p*-Hydroxybenzoic acid	56.0 ± 3.4
	Syringic acid	37.9 ± 7.6
	Vanillic acid	58.6 ± 17.6
Hydroxycinnamic acids	*p*-Coumaric acid	126 ± 11.0
	Ferulic acid	54.9 ± 5.5
	Caffeic acid	57.8 ± 4.0
	Cinnamic acid	28.0 ± 5.6
	Sinapic acid	11.9 ± 1.2
Chlorogenic acids	5-*O*-caffeoylquinic acid	280 ± 14.0
Flavan-3-ol-derivates	Epigallocatechin gallate	135 ± 13.0
Flavan-3-ols	Catechin	115 ± 12.0
	Epicatechin	158 ± 16.0
**Flavonol-glycosides**	**Quercetin-3-*O*-glucoside**	**3814 ± 114.0**
	Quercetin-3-*O*-L-rhamnoside	358 ± 21.0
	**Quercetin-3-*O*-galactoside**	**2067 ± 124.0**
	**Kaempferol-3-*O*-glucoside**	**4182 ± 167.0**
	Quercetin-3-*O*-rutinoside	389 ± 12.0
Flavone glycosides	Luteolin-7-*O*-glucoside	30.9 ± 0.9
	Vitexin	0.73 ± 0.0
	Apigenin-7-*O*-glucoside	11.4 ± 0.6
**Flavonols**	Myricetin	47.6 ± 4.8
	**Quercetin**	**2341 ± 702.0**
	**Kaempferol**	**1009 ± 71.0**
	Isorhamnetin	44.6 ± 2.7
Flavanones	Naringenin	46.9 ± 3.3
Flavones	Luteolin	292 ± 15.0
	Apigenin	23.4 ± 1.6
	Chrysoeriol	10.2 ± 0.3
Coumarins	Esculetin	4.53 ± 0.3
	Scopoletin	20.0 ± 1.6

* The results are expressed as the concentration (µg/g of dry extract) ± the relative standard deviation of repeatability, as determined by method validation.

**Table 3 nutrients-16-03368-t003:** Viable counts (log CFU/mL) of selected probiotic strains incubated with spent PH material.

Psychobiotic Strain	Viability (log CFU/mL)			
	C		Spent PH Material	
	0 h	4 h	0 h	4 h
*Lactiplantibacillus plantarum* 299V	7.93 ± 0.32	9.02 ± 0.54	8.17 ± 0.34	9.26 ± 0.54
*Limosilactobacillus reuteri* DSM 17938	7.80 ± 0.38	8.45 ± 0.07	8.38 ± 0.05	8.88 ± 0.97
*Heyndrickxia coagulans* (formerly known as *Bacillus coagulans)*	8.22 ± 0.30	8.74 ± 0.31	8.36 ± 0.26	8.99 ± 0.82
*Escherichia coli* Nissle 1917	6.00 ± 0.61	8.18 ± 0.67	6.32 ± 0.06	8.21 ± 0.73

## Data Availability

The original contributions presented in this study are included in the article/Appendix A; further inquiries can be directed to the corresponding authors.

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
