# Peer review of "New Perspectives on the Old Uses of Traditional Medicinal and Edible Herbs: Extract and Spent Material of Persicaria hydropiper (L.) Delarbre"

_nutrients, 2024, doi:10.3390/nu16193368_

Round 1

Reviewer 1 Report

Comments and Suggestions for Authors

Extract and Spent Material of Persicaria hydropiper: The Combination of Zero Waste and Flexitarian Nutrition

linha 19-20- O titulo deveria ser revisto em virtude do objetivo proposto

ABSTRACT

Line 18-33- The article does not clearly present the methodological bases applied (Adjust).

The article does not mention any research on the functional ingredient of old traditional herbs...item 2.2 (A Survey Regarding the Functional Ingredient from Old Traditional Herbs and Spent Plant Material). (Adjust)

INTRODUCTION

Line 59- Add more specific references on the topic

Line 84- If the literature is extensive ..... Then add more sources of references to support your statements

Line 86- I suggest including references 22 and 23 to better support the update (recent publications)

Line 90- I suggest including more current references.

Line 92- I suggest including more current references.

Line 100- The objective should be standardized as presented in Lines 19-20 of the abstract

METHODOLOGY

Line 133 - item does not present a direct relationship with the objectives proposed in the research, expressed in Lines 19-20 and partially resumed in line 100.

RESULTS AND DISCUSSIONS

Line 267 - item 3.1. I suggest reviewing the presence of item 3.1 as it does not present a direct relationship with the research theme proposed in this article (lines 267 to 328) in addition to providing data that should be exclusively in the methodology item 2.2

I suggest reviewing the proposed objectives or removing this item

Line 329-336 item 3.2 - I suggest reviewing and being more objective in the presentation of the results of Lines 330 to 336, as they are not in accordance with the presentation of the data. Line 355-357 ... if there are several studies, more than one should be cited.

Line 376-381 - the preamble presented in the excerpt is out of context for the topic, which should be more objective and present only the results and discussions.

Line 387-388 - The numerous studies should be cited.

Line 409 - highlight the safety of experiments at this stage, supported by only 1 study (Line 408). The statement (Line 410) is generalist and should be supported by more references, with more advanced stages in humans based on international government agencies or entities (e.g. WHO, FDA, etc.)

Line 416 - The studies cited refer to the effects related to the solvent used in the preparation of the extract... With a wide range of solvents to choose from for preparing pure or mixed extracts, how can the action/effects of the solvents be mitigated? Lines 439-441 - This data should only be in the methodology

Lines 443-447 - The comparison with such different raw materials does not seem appropriate, there is a need for references with the same aspect or botanical family.

Lines 453-455 - the presented excerpt should be placed in the appropriate item, that is, in item 4 Conclusions

CONCLUSIONS

Lines 463-475-I suggest completely reviewing the construction of this paragraph, since it does not clearly respond to the proposed objectives. In addition, it expresses a great deal of value judgment and a high potential for generalization in solving serious problems related to food based on a single source of unconventional foods.

Reviewer 2 Report

Comments and Suggestions for Authors

The manuscript entitled “ Extract and Spent Material of Persicaria hydropiper: The Combination of Zero Waste and Flexitarian Nutrition”

  The work is generally interesting, especially in the context of interest in the health-promoting properties of various plants traditionally used in folk medicine. However, I do not quite understand whether the zero waste trend applies in this case? What is the scale of the problem? How much of the spent plant material Persicaria hydropiper is thrown away? Such information should be included in the introduction.

Secondly, what was actually done with the residue after extraction/maceration of the plant powder? The authors state that they dried it and treated it as spent material. Ok. Only in all subsequent experiments, they used the plant extract (after evaporation of the solvent dissolved in DMSO). Nowhere in the materials and methods is it written that this spent material was tested? All analyses concern PH - LC/MS, cytotoxicity, viability of A. saline and the effect on bacteria (line 186). And in the results (Table 3) there is spent PH material. This should be explained in detail and appropriate information should be added in the material and methods section. The results should also clearly state whether this applies to extract or spent.

 In Table 3 - was the effect of spent material PH on bacterial viability statistically significant? There is no information in the description of the results or under the table. If the differences are not statistically significant, it is unlikely to be stated that it has a beneficial effect on the growth of probiotics.

 In addition, in the description of the cytotoxicity analysis and the effect on A. saline, it should be stated what constituted the control solution. Did the control contain the same amounts of DMSO as the test samples? Or was the amount of DMSO so low that there is a certainty that the solvent itself did not affect the viability of the cells or the test organism?

 Do the authors have data and/or can determine what PH concentrations are actually safe for the consumer? And how does this translate into the amount of PH herb or infusion/extract consumed? What is the order of magnitude - a glass of infusion, two per day, 10 grams of fresh leaves in a dish?
